# Silk Protein Composite Bioinks and Their 3D Scaffolds and In Vitro Characterization

**DOI:** 10.3390/ijms23020910

**Published:** 2022-01-14

**Authors:** Ji-Xin Li, Shu-Xiang Zhao, Yu-Qing Zhang

**Affiliations:** School of Biology and Basic Medical Sciences, Medical College, Soochow University, RM702-2303, No. 199 Renai Road, Industrial Park, Suzhou 215123, China; 20205221002@stu.suda.edu.cn (J.-X.L.); 20184221050@stu.suda.edu.cn (S.-X.Z.)

**Keywords:** silk protein, fibroin, sericin, grafting, composite bioink, bioprinting, 3D scaffolds, mechanical properties

## Abstract

This paper describes the use of silk protein, including fibroin and sericin, from an alkaline solution of Ca(OH)_2_ for the clean degumming of silk, which is neutralized by sulfuric acid to create calcium salt precipitation. The whole sericin (WS) can not only be recycled, but completely degummed silk fibroin (SF) is also obtained in this process. The inner layers of sericin (ILS) were also prepared from the degummed silk in boiling water by 120 °C water treatment. When the three silk proteins (SPs) were individually grafted with glycidyl methacrylate (GMA), three grafted silk proteins (G-SF, G-WS, G-ILS) were obtained. After adding I2959 (a photoinitiator), the SP bioinks were prepared with phosphate buffer (PBS) and subsequently bioprinted into various SP scaffolds with a 3D network structure. The compressive strength of the SF/ILS (20%) scaffold added to G-ILS was 45% higher than that of the SF scaffold alone. The thermal decomposition temperatures of the SF/WS (10%) and SF/ILS (20%) scaffolds, mainly composed of a β-sheet structures, were 3 °C and 2 °C higher than that of the SF scaffold alone, respectively. The swelling properties and resistance to protease hydrolysis of the SP scaffolds containing sericin were improved. The bovine insulin release rates reached 61% and 56% after 5 days. The L929 cells adhered, stretched, and proliferated well on the SP composite scaffold. Thus, the SP bioinks obtained could be used to print different types of SP composite scaffolds adapted to a variety of applications, including cells, drugs, tissues, etc. The techniques described here provide potential new applications for the recycling and utilization of sericin, which is a waste product of silk processing.

## 1. Background

To function as a biological ink, biological material must be cell-friendly, cell-compatible, have strong mechanical properties, and be able to adhere to cells under physiological conditions. During the process of 3D printing, the printability of bioink mainly depends on its surface tension, swelling rate, and viscosity. The surface tension of bioink has a strong influence on cell attachment, distribution, and development in 3D structures [1]. The swelling rate affects the formation of the two-dimensional morphology of the bioink after extrusion, and the appropriate swelling rate will improve the resolution of bioprinting products. When viscosity is high, a greater extrusion force is needed during the 3D printing process of bioink, which could block the nozzle, as well as damage cell vitality [2,3,4]. In addition, biomaterials used as bioink should be inexpensive, easy to obtain, and simple to manufacture.

Silk derived from the silkworm *Bombyx mori* is a natural protein-based polymer composed of approximately 70–80% silk fibroin (SF) and ~20–30% silk sericin [5]. The silk fibroin has a molecular mass of more than 410 kDa and consists of a heavy chain (~390 kDa) and light chain (~25 kDa) [6,7]. The heavy chain has a GAGAGS repeat sequence accounting for ~70–80% of the total [8]. These motifs constitute the crystallinity domain of SF, while other amino acids with larger side chains constitute the amorphous domain [9,10]. Due to its excellent mechanical properties [11] and biocompatibility, with no immunogenicity or toxic side effects, SF fiber has received attention recently in research on the application and development of biomaterials, for example, dye adsorption material [12], antibacterial material [13], conductive material [14,15], and especially materials that are suitable for medical tissue engineering, drug-carrying composite materials [16], and bone tissue engineering scaffold [17].

When the silk fiber is degummed in boiling or alkaline water, the regenerated silk fibroin, with or without [18] chemical modification, can be made into a starting material for the development of 3D bio-printing ink [19,20]. SF can also be blended or combined with other polymers such as alginate, chitosan, and gelatin to alter the rheological properties of the bioink and significantly improve its printability, gelation rate [21], and cell growth [22], as well as enhancing its mechanical properties and reducing the degradation rate [23,24]. When mixed with hydroxyapatite, the SF membrane can enhance mechanical properties and promote cell proliferation and has applications in wound healing or as printable or injectable bone filling material [25]; silk fibroin and bioactive glass hybrid 3D printed composite scaffold for bone tissue engineering [26]; and silk fibroin and I_3_K peptide nanofiber scaffold 3D printed cell scaffold [27]. It can also induce silk fibroin–hyaluronic acid to generate hydrogel by ultrasound [28]. After covalent crosslinking of the SF protein using riboflavin or horseradish peroxidase, the structure of the photocurable SF hydrogel is composed of β-folding, α-helix, and random coil structures [29], which could promote the attachment and growth of mouse fibroblasts. SF modified by methacrylic anhydride (MA) or glycidyl methacrylate (GMA) in the presence of photoinitiators can promote the curing of silk hydrogels, as long as they are irradiated by UV light, and can be printed into various highly complex organ structures with excellent mechanical and rheological properties [29,30]. The reaction occurs by introducing GMA into the reactive side chain (–NH_2_, –OH, –COOH) of SF. The graft modification of silk fibroin and GMA for 3D printing has a wide range of applications including digital light processing for cartilage tissue engineering, and 3D printing GMA cross-linked silk fibroin hydrogel [31]. GMA cross-linked silk fibroin is used in the field of tissue engineering [32]. GMA grafted silk fibroin is used for meniscus tissue engineering [33].

As mentioned above, the degumming process of silk fibers will produce degummed SF, as well as another silk protein, sericin. The traditional degumming process yields SF fibers, while sericin is often lost along with large amounts of alkaline waste. Therefore, the recycling and utilization of sericin has long been a focus of the silk processing industry.

Sericin is a globular protein with a molecular weight of 14 to 314 kDa, which surrounds the SF fibers. Like SF, it is composed of eighteen amino acids, but their composition ratios differ substantially. Most of them are primarily composed of polar amino acids. Sericin has many properties including antioxidant activity, bacteriostasis, anti-UV and anti-inflammation activity, wound healing, whitening (inhibition activity against polyphenol oxidase), and blood sugar reduction [34,35,36,37,38]. Sericin is rich in water-soluble amino acids, has anti-oxidation and anti-tyrosinase properties, can inhibit polyphenol oxidase activity, and contains active substances such as flavonoids. Sericin has been used as an additive for cosmetics, skin and haircare products, cell culture media, serum substitutes for cell culture, and as a cryoprotectant [39,40,41,42,43]. In terms of structure, sericin is wrapped around the silk fibroin fiber and is distributed in layers. Molecular biology studies have shown that there are several layers of sericin with different amino acid compositions and molecular masses [44,45]. So far there has been no way to separate and purify these components individually. The authors have prepared three layers of sericin peptides with different molecular weights by different degumming methods. The outer, middle, and inner layers of sericin account for 15%, 10.5%, and 4.5% of the total weight, respectively [46]. The amino acid composition of each sericin layer is also substantially different. The outer layer of sericin has good water solubility and a large molecular weight, while the inner layers of sericin are poorly soluble in water. Sericin and methacrylic anhydride are photocrosslinked to form an injectable hydrogel [47]; 3D printable hybrid hydrogels are composed of sericin and methacrylic anhydride modified gelatin (GelMA) [48]. At present, there is no report about mixing grafted silk fibroin and sericin for 3D printing.

The aim of the present study was to focus on the synthesis and preparation of silk protein composite bioinks, including SF, WS (whole sericin), and ILS (inner layers of sericin). The main purpose of our investigation was to improve the biological performance of a single silk fibroin bioink by adding whole sericin and inner sericin to the bioink’s composition. We used a green degumming method, using an aqueous solution of hydrated calcium hydroxide (Ca [OH]_2_) to degum the silk or cocoon shell, not only as a process to obtain SF, but also to recover WS [49]. The ILS was prepared using water at high temperatures and pressures. Following the grafting reaction, the grafted silk proteins were bioprinted into 3D silk protein composite scaffolds, and their mechanical and structural properties as well as their biological characteristics and drug-release potential were systematically investigated.

## 2. Results

### 2.1. Bioink Synthesis

#### 2.1.1. Synthesis of Biological Ink

The pure grafted SF (G-SF) bioink prepared with single-component GMA-grafted SF was relatively clear and transparent (Figure 1 and Table 1), and the WS or ILS percentages increased with the addition of G-WS (2–4) or G-ILS (5–7), which resulted in a darkening in the color of the composite bio-ink. Moreover, the color of the composite bioink with G-WS was darker than that with G-ILS, which is likely because the color of whole sericin powder is darker than that of inner sericin powder [50].

#### 2.1.2. Three-Dimensional Biological Printing

See Figure 2 for the porous biological scaffold prepared by 3D bioprinting of the bioink or composite bioink. As shown in Figure 2a,b, the biological scaffolds based on silk fibroin and sericin are printed with the words “Soochow University”, and various biological scaffolds can clearly be identified by color, which shows that they are translucent, soft, and elastic. Thus, biological scaffolds based on silk fibroin and sericin demonstrate shape plasticity, which can be adapted for a variety of medical applications.

### 2.2. Mechanical Properties

#### 2.2.1. Effect of GMA Content on Mechanical Properties of SF-Alone Scaffold

As shown in Figure 3a, the stress of the G_1.0_-SF scaffold at 75% strain was 0.145 MPa, while the maximum compressive strains of the G_2.5_-SF, G_5.0_-SF, and G_7.5_-SF scaffolds were 74%, 71%, and 66%, respectively, and the compressive strengths were 0.18 MPa, 0.22 MPa, and 0.22 MPa, respectively. It can be seen that the higher the degree of GMA substitution, the greater the compressive strength of the scaffold and the smaller the maximum compressive strain. Figure 3b shows the compressive modulus of the four biological scaffolds by calculating the slope of the linear elastic region (below 5%) in the stress–strain curve. Among them, the average compressive modulus of the G2.5-SF scaffold was 0.033 MPa, and the G7.5-SF scaffold had the highest compressive modulus. However, the compressive modulus of the G_1.0_-SF bioscaffold was only about 0.022 MPa, which was much smaller than the compressive modulus of the other three and could not meet the mechanical properties required by biomedical materials. The results showed also that the amount of methacrylic acid anhydride would affect the mechanical properties of the bioscaffold [51]. Since the maximum compressive strain of the G_2.5_-SF bioscaffold was higher than that of the G_5.0_-SF and G_7.5_-SF scaffolds, the following experiments used a 2.5% GMA level in all SP scaffolds for silk protein grafting. Fresh samples were used for each test.

#### 2.2.2. Effect of Silk Degumming Agent Concentration on Mechanical Properties of Pure SF Scaffold

Figure 4a shows the three pure SF scaffolds synthesized by three SFs degummed in 0.05%, 0.075%, and 0.1% Ca(OH)_2_ aqueous solutions for boiling treatment (40 min), respectively. Their maximum compressive strains were 74%, 80%, and 84%, and their compressive strengths were 0.18, 0.177, and 0.15 MPa, respectively. It can be seen that with the increase of Ca(OH)_2_ concentration, the compressive strength of the pure SF scaffold decreased to varying degrees. These results indicate that a high concentration of Ca(OH)_2_ in water will partially cause the fracture of the SF peptide chain and eventually damage the mechanical properties of the grafted SF scaffold. Moreover, as shown in Figure 4b, the compression modulus of the SF scaffold obtained by the lowest concentration of 0.05% Ca(OH)_2_ degumming was the highest (mean value, 0.034 MPa) of all. Therefore, the higher the concentration of Ca(OH)_2_, the smaller the compressive modulus of the SF scaffolds. Our previous reports indicated that the polypeptide chain of SF degummed with 0.025% Ca(OH)_2_ was still intact [49]. Therefore, in the following experiments, SF fibers degummed with 0.05% Ca(OH)_2_ were selected to prepare SF bioink (G-SF).

#### 2.2.3. Effect of Silk Degumming Time on Mechanical Properties of Pure SF Scaffold

After using 0.05% Ca(OH)_2_ to degum the cocoon shells for 10, 20, 30, 40, and 50 min, respectively, the mechanical properties of the pure SF scaffold made from degummed SF were measured. As shown in Table 2, the compressive strengths of the pure SF scaffolds obtained from degumming for 10, 20, 30, and 40 min were almost the same, while the compressive strength of the SF scaffolds obtained from degumming for 50 min was only 0.17 MPa. Moreover, with the increase in degumming time, the maximum compressive strain and compressive modulus of the biological scaffold showed a negative correlation trend. The compressive modulus of the pure SF scaffold obtained by degumming for 10 min was 0.055 MPa, which is the highest value of all degumming times from 10 to 50 min. This result was the same as the above-mentioned Ca(OH)_2_ degumming concentration. Longer degumming times resulted in increased breakage of the silk fibroin peptide chain, such that the resulting pure silk fibroin scaffold demonstrated worse mechanical properties. Therefore, in the following experiments, the silk fibroin that had undergone 0.05% Ca(OH)_2_ degumming for 10 min was grafted with 2.5% (*v*/*v*) glycidyl methacrylate (G) to prepare the SP bioinks.

#### 2.2.4. Effect of Incorporating Grafted Sericin on the Mechanical Properties of SP Scaffolds

As shown in Table 3, as the concentration of the two grafted sericins increased, the mechanical properties of the composite scaffolds changed compared with SF alone. Among them, the compressive modulus of SP composite scaffolds added with G-WS was significantly reduced, while the maximum compressive strain was significantly increased. The compression modulus and the maximum compressive strain of the SF scaffold were ~0.055 MPa and ~70%, respectively. The compression modulus of the SF/WS (30%) scaffold showed a significant decrease, which was only one third of that of the SF alone. The compressive strain increased by about 6%. The compressive strength of SP composite scaffolds with 10% and 20% G-WS was similar to that of pure SF scaffold, while the compressive strength of SP scaffolds with 30% WS was significantly inferior, about half of that of the pure SF scaffold. However, the compressive strength and maximum compressive strain of the SP composite scaffolds with the addition of G-ILS increased to varying degrees, but the compressive modulus decreased significantly. The compressive strength and compressive modulus of the SF/ILS (20%) scaffold reached 0.29 MPa and 0.042 MPa, respectively. Thus, the addition of 10% WS or 20% ILS significantly improved the compressive strength and maximum compressive strain of the SP composite scaffold. Therefore, we selected the SF/WS (10%) and SF/ILS (20%) composite scaffolds for the following experiments.

### 2.3. Microstructure

It can be seen from Figure 5 that all SP scaffolds have a good porous network structure, which provides optimal properties for cell adhesion and migration. As shown in Table 4, the average pore diameter of the pure SF scaffold was 70.7 ± 18.2 µm, while the average pore size of the SP composite scaffolds SF/WS (10%) and SF/ILS (20%) added with G-WS and G-ILS had increased slightly to ~71.2 µm and ~74.9 µm, respectively. When a porous scaffold is used for skin tissue repair, the ideal pore size is 20–125 µm [52]. Thus, the aforementioned SP scaffolds meets the cell-growth pore size requirements for skin tissue repair. The observed porosity of each group of scaffolds is shown in Table 4. The porosity of the pure SP scaffold was 79.7 ± 4.6%, while the porosity of the modified SP composite scaffolds SF/WS (10%) and SF/ILS (20%) was slightly larger. Therefore, whether the whole sericin or the inner sericin was incorporated, it had little effect on the average pore size and porosity of the SP composite scaffold. This porous scaffold is very suitable for cell growth and can facilitate the exchange of nutrients between the cells and the extracellular environment.

### 2.4. Infrared Spectrum

As shown in Figure 6 silk fibroin, whole sericin, inner sericin, and silk protein scaffolds all showed two distinct characteristic peaks between 2000 cm^−1^ and 1000 cm^−1^. Among them, the first peak appeared at about 1650 cm^−1^, the absorption peak of the α-helix of amide I was in the range of 1650–1658 cm^−1^, with the absorption peak in the range of 1640–1648 cm^−1^, indicating a random coil and α-helix structure in amide I [50,53]. The absorption peak of β-sheet in amide II was in the range of 1515–1525 cm^−1^, the absorption peak of random coil was in the range of 1535–1545 cm^−1^, and the second peak of silk fibroin and sericin appeared at 1539 cm^−1^ and 1537 m^−1^, with both belonging to the random coiled structure. The second peaks of the ILS, pure SF, and SP composite scaffolds were at about 1519 cm^−1^, which belong to the β-sheet structure. This shows that after the SF is grafted and solidified into a 3D scaffold, its structure has rearranged from random coils to a β-sheet structure. SF, WS, ILS, and SP scaffolds all had a small peak at about 1235 cm^−1^, corresponding to the random coil structure in amide III. In addition, the characteristic peaks of pure SF, SF/WS (10%), and SF/ILS (20%) scaffolds were almost the same, indicating that the addition of G-WS or G-ILS did not change the secondary structure and performance of these SP composite scaffolds.

SF, WS, and ILS were freeze-dried powder samples of silk fibroin, whole sericin, and inner silk sericin, respectively; pure SF, SF/WS (10%), and SF/ILS (20%) represent grafted SF alone scaffold and two SP composite scaffolds mixed with 10% G-WS and 20% G-ILS, respectively.

### 2.5. X-ray Diffraction

Silk fibroin has two crystal structures: silk I and silk II. The main conformation of silk II consists of β-sheets. In the past, it was generally considered that silk I has an α-helical structure. It is now generally accepted that silk I is a type II β-turn [54]. As shown in Figure 7, the powdered SF sample had less β-folded structure, and its diffraction peak was at ~20°. The pure SF scaffold had two typical characteristic peaks belonging to silk II at 20.7° and 24°. When the silk fibroin was grafted and solidified into a biological scaffold, the content of the β-sheet structure was significantly increased. Whole sericin had a wider peak, which is the amorphous part, and the inner sericin had a strong peak at 19.2°, which belongs to the crystalline part. Both the pure SF scaffold and the SF composite scaffold had broad peaks at 20.7° and 24°, corresponding to the silk II structure, which indicates that the SP composite scaffold incorporating G-WS or G-ILS still contains a β-sheet structure. This result is consistent with the above FT-IR result.

SF, WS, and ILS were freeze-dried powder samples of silk fibroin, whole silk fibroin, and inner silk fibroin, respectively; SF alone, SF/WS (10%), and SF/ILS (20%) scaffolds represent grafted SF scaffolds and two SP composite scaffolds mixed with 10% G-WS and 20% G-ILS, respectively.

### 2.6. Thermal Analysis

From Figure 8a, it can be deduced that the thermal degradation of SF, WS, ILS, and SP scaffold is divided into four stages, which are characterized by a significant mass loss rate. Among them, the initial weight loss of the WS and ILS was detected below 80 °C and 90 °C, respectively, and the initial weight loss was detected for both SF and SP scaffolds below 120 °C. This was the first stage, mainly caused by evaporation of water in the sample. At that point, the WS and ILS were between 80 and 200 °C and 90 and 210 °C, respectively, and both SF and SF scaffolds showed low mass loss between 120 and 240 °C. This was the second stage, perhaps induced by loss of some degraded silk protein peptides with low molecular mass in the sample. Then, in the third stage, the WS and ILS were between 200 and 460 °C and 210 and 460 °C, respectively, and the decomposition rate of SF and the SF scaffold accelerated sharply at 240–460 °C. The mass was quickly lost, with the lost mass representing about 55% of the total mass of the sample. Finally, the SF, WS, ILS, and SP scaffolds gradually stabilized after the fourth stage at 460 °C.

SF, WS, ILS silk fibroin, whole silk fibroin, and inner silk fibroin powder; and SF alone, SF/WS (10%), and SF/ILS (20%) represent grafted SF scaffolds and two SP composite scaffolds mixed with 10% G-WS and 20% G-ILS, respectively.

It can be seen from Figure 8b that the degradation process of SF, WS, and ILS show maximum values around 286 °C, 324 °C, and 314 °C, respectively. At that time, the peptide bond inside the SP molecule was severely broken, the C–N bond with a lower bond energy was the most severely broken, and the C–C bond, C=O bond, and N–H bond were also broken. The maximum thermal decomposition of the SF scaffold and SP composite scaffolds of pure SF, SF/WS (10%), and SF/ILS (20%) was around 291 °C, 294 °C, and 293 °C, respectively, which indicates that degradation of the three composite scaffolds required more energy than SF alone, and the thermal stability was greater. These results suggest that GMA modification significantly improves the thermal stability of the SP composite scaffolds.

### 2.7. Swelling Rate

The water absorption capacity of the scaffold is a primary factor in the hydrolysis of the scaffold and for the diffusion of culture medium and nutrients. The swelling curves of SP scaffolds are shown in Figure 9. The swelling properties of bioscaffolds are an important index for evaluating their water absorption capacity. A high-swelling scaffold is a consequence of its porous structure and the number of hydrophilic groups.

As shown in Figure 10, for the first 6 h, the swelling rate of all SP composite scaffolds increased substantially and gradually reached the swelling equilibrium state after 6 h. After 12 h of expansion, the swelling ratio of the final SF/WS (10%) and SF/ILS (20%) SP scaffolds was about 2.7 times the initial weight and were higher than the swelling rate of the pure SF scaffold. This indicates that the swelling property of SP composite scaffolds could be improved by adding WS and ILS.

### 2.8. Enzyme Degradation

In this experiment, the neutral protease hydrolysate silk protein scaffold was used. The enzymatic hydrolysis curve is shown in Figure 10. The mass residual rate of the pure SF scaffold 20 days after enzymatic hydrolysis was still about 38%, while G-WS and G-ILS added to both SP composite scaffolds resulted in rate of about 23% and 25%, respectively. This indicates that the addition of the two GMA-modified sericin SP composite scaffolds was easier to degrade, possibly due to the easier degradation of sericin. This suggests that the structure and properties of GMA-sericin-incorporated into SP composite scaffolds may be more suitable for various applications in medical tissue engineering materials.

### 2.9. Drug Delivery Capacity

In this paper, bovine insulin was selected as a model drug compound to investigate the effect of its release on SP composite scaffolds. Insulin was encapsulated in a protein scaffold, and the release histogram is shown in Figure 11.

The insulin release histogram of the pure SF scaffold is shown in Figure 11. On the first day, the insulin release rate of the SF scaffold and SP composite scaffolds were both approximately 20%. With the extension of the release time, the release rate of pure SF scaffolds and SF/WS (10%) and SF/ILS (20%) SP scaffolds gradually increased, until the release rate after the final 5 days reached about 52%, 61%, and 56%, respectively. Among them, the release rate of the SF/WS (10%) scaffold from day 2 to day 5 was higher than that of the other two groups. From the overall trend, the insulin release rate of these three SP scaffolds increased linearly within 5 days, regardless of whether the WS or ILS was incorporated, ultimately reaching 52–62%. The results show that the addition of G-WS and G-ILS could increase the insulin release efficiency of the SP composite scaffold, and the drug release performance of the SF/ILS (20%) scaffold was similar to that of the pure SF scaffold.

### 2.10. Cytocompatibility

Using L-929 mouse fibroblasts as model cells in this experiment, the cells were inoculated with the same cell density on the surface of sterile, treated SP composite scaffolds. The cell growth status was observed and recorded regularly, and the cell viability of L-929 on various scaffolds was measured on days 1, 2, 3, and 4 (Figure 12).

From day 1 to 4 of the inoculated cells, the growth state of the cells on the three groups of L-929 scaffolds was observed under normal light and fluorescence, respectively. The results are shown in Figure 12. After a day of inoculation on the surface of the protein scaffold, most of the cells were round and sparse and evenly distributed on the surface of the bioscaffold, all of them gradually adhered to the wall and attached, and the cell morphology changed from spherical to fusiform to more clearly observed changes in cell growth and morphology and eliminated background effects and distinguished between living and dead cells. L-929 was stained by fluorescence using the AM of calcitriol and propidium iodide PI. On the 4th day, the cells proliferated on the surface of the scaffolds, and almost all of them were fusiform. The pseudopods of the cells were elongated, mostly showing mature fusiform morphology, and the cells grew vigorously, while the red (dead) cells were few and almost invisible. The SP composite scaffolds promoted support and adhesion of L-929, as well as proliferation.

In order to further evaluate the impact of bioprinting composite scaffolds on cell growth and adhesion, CCK-8 was used to measure cell viability on various SP scaffolds every day for 4 days. As shown in Figure 12d, the three groups of pure SF scaffold and SP scaffolds did not have significant differences in cell division and growth on the first day, but the SP composite scaffolds with added sericin seemed to grow slightly better on the second and third days. By the fourth day, the SP composite scaffold SF/ILS (20%) mixed with inner sericin showed the best growth and cell viability among the three groups. In general, these three SP scaffolds demonstrated good cell compatibility with L929 cells.

## 3. Discussion

As a by-product of the silk textile industry, sericin is widely available, though it is often regarded as a waste product of traditional silk processing. The side chain of sericin contains more –NH_2_ and –COOH functional groups than that of silk fibroin, which facilitates the synthesis of new products through chemical crosslinking. However, there are few reports on the applications of sericin in the preparation of bioinks. Sericin, especially partially degraded sericin, has good biocompatibility and biodegradability, comparable to that of silk fibroin. It has excellent biological properties, including antioxidant, anti-inflammatory, and antibacterial activity, as well as whitening effects and glucosidase inhibition [55]. Sericin has also been found to promote cell adhesion, growth, and migration [56]. Therefore, partially degraded sericin was incorporated into the SF scaffold base of an SF bioink in order to ensure that the SP composite scaffolds not only possessed certain mechanical properties, but also properties that promoted cell adhesion, growth, and proliferation.

The SF bioink printing scaffolds and silk fibroin composite scaffolds have certain mechanical properties, as well as a porous network structure, which are conducive to the transportation of bioactive substances and the discharge of cell metabolites. Furthermore, they improve cell adhesion and growth, an important requirement for medical tissue engineering. For example, SF, WS, and ILS bioinks can be bioprinted into corresponding SP scaffolds by using single ink or composite inks in accordance with specific medical application requirements. After the SF ink is mixed with the relevant cells, the holes removed from bone tumors, for example, could be filled by injection, followed by the introduction of ultraviolet radiation through optical fibers to fix the bioink and accelerate the proliferation and repair of cells. SF could be used as a wound dressing or artificial skin to repair tissue, along with the use of related growth factors and anti-inflammatory drugs, rather than a photoinitiator.

In addition, traditional oral or injected medications typically result in short-term drug concentrations in human blood that are substantially higher than that needed for treatment, which may increase side effects or lead to a decrease in efficacy. SP scaffolds have potential applications as drug carriers. An SP scaffold could release drugs gradually through diffusion and penetration, thereby prolonging the efficacy and flexibly controlling the drug release site. Scaffolds should be biocompatible, so as to integrate with the tissue around the implant site in order to avoid rejection. Sericin and silk fibroin are natural animal proteins, and biological scaffolds will be gradually degraded by proteases in vivo. The degradation products are small peptides or amino acids, which have no toxic effects and can be absorbed and utilized by the human body. This process could provide suitable space for new tissue and release the relevant growth factors. Moreover, degradation of the materials in vivo would mean that no additional surgery would be required, which would provide an additional benefit to patients. Some studies have shown that during the degradation process, sericin shows increased antioxidant activity and thereby can inhibit cellular damage due to oxidative stress damage as well as apoptosis. Therefore, silk protein scaffolding implanted in vivo, even while undergoing degradation, not only shows higher antioxidant activity, but also promotes cell adhesion and growth, in addition to cell proliferation, among other characteristics. The use of silk protein scaffolds as a novel medical biomaterial provides a starting point for exploring potential applications for sericin and silk fibroin. However, in vivo verification of the efficacy of silk fibroin scaffolds needs further research, and the scope of potential applications requires further discussion.

## 4. Materials and Methods

### 4.1. Materials

Silkworm *Bombyx mori* cocoons were purchased from Nantong New Silk Road, Jiangsu Province (produced in spring 2019). Silkworm varieties were “Su Hao × Zhong Ye “; mouse fibroblasts were L-929, purchased from Punosai, Wuhan, China. Glycidyl methacrylate (GMA) and neutral protease (50 U/mg) were purchased from Shanghai Aladdin (Shanghai, China), and the photoinitiator Irgacure 2959 (2-hydroxy-4’-(2-hydroxyethoxy)-2-methylpropiophenone) was purchased from Shanghai Saen (Shanghai, China).

### 4.2. Separation and Purification

The cocoon shell was immersed in 60 vol of distilled water and boiled for 2 h to remove the outer layers of sericin. To extract the ILS, the degummed silk fiber was treated with 120 °C water under high pressure for 2 h. The degummed solution containing ILS was concentrated, filtered, and spray-dried at 110 °C to obtain powdered ILS.

In addition, a saturated aqueous solution of Ca(OH)_2_ was used as a green degumming agent to prepare a fibrous silk fibroin (SF) without sericin and a partially degraded sample powder of WS [42]. The alkaline solution of Ca(OH)_2_ is used as a silk degumming agent, not only to obtain a fibrous silk fibroin (SF) without sericin, but also to obtain an alkaline sericin solution containing Ca^++^ ions, which is neutralized by sulfuric acid or phosphoric acid [42]. WS is obtained by removing the precipitated calcium salts.

### 4.3. Grafting Methods

(1)The dried SF fiber obtained from the Ca(OH)_2_-degumming process described above was dissolved in a water bath (60 °C) with a 9.3 M solution of LiBr solution and agitated to obtain the SF salt solution (10%, *w*/*v*). Next, the SF salt solution was placed on a magnetic stirrer, and the following concentrations of GMA were added dropwise to the 100 mL SF salt solution (10%, *w*/*v*), respectively, at a speed of approximately 0.5 mL/min: 1.0 mL, 2.5 mL, 5.0 mL, and 7.5 mL. The grafting conditions were as follows: rotating speed, 300 rpm; temperature, 60 °C; time, 3 h. The grafted solution was transferred to a dialysis bag (MW: 14,000) and dialyzed for 3 days, during which the deionized water was changed every 6 h. Finally, the grafted silk fibroin (GMA-SF, abbreviated to G-SF) was frozen at –80 °C, and then freeze-dried into a porous foam structure and stored in the dark. The four types of grafted silk fibroin were designated as G_1.0_-SF, G_2.5_-SF, G_5.0_-SF, and G_7.5_-SF, respectively.(2)The powdered WS and ILS obtained above were dissolved in deionized water to give a 10% (*w*/*v*) sericin solution. Next, the solution was placed on a magnetic stirrer, and the same amount of G-SF described above was added dropwise at a speed of about 0.5 mL/min. A dialysis bag (MW: 3500) was used for the dialysis. The remaining steps were the same as those for making G-SF. Two types of grafted sericin were obtained, namely grafted whole sericin (GMA-WS, abbreviated as G-WS) and grafted inner sericin (GMA-ILS, abbreviated as G-ILS), and the samples were kept away from light.

### 4.4. Biological Ink Formulation

An accurately weighed sample of 0.1 g I2959 was dissolved in 10 mL PBS (phosphate buffer). The following amounts of G-SF or G-ILS were subsequently added in sequence, respectively, namely 5.0 g, 4.5 g, 4.0 g, and 3.5 g of G-SF, and 0, 0.5 g, 1.0 g, 1.5 g of G-ILS, then stirred until dissolved. PBS was added dropwise to increase the volume to 20 mL. The final concentration of I2959 was 5% (*w*/*v*), and seven different bioinks of 25% (*w*/*v*) were obtained and designated as follows: G-SF, G-SF/WS (10%), G-SF/WS (20%), G-SF/WS (30%), G-SF/ILS (10%), G-SF/ILS (20%), and G-SF/ILS (30%).

### 4.5. Testing of Mechanical Properties

The 3D scaffolds were cylindrical in shape with a diameter of 12 mm and a height of 8 mm. The compression properties of the scaffolds were determined with an electronic tensile testing machine (WH-5000, Ningbo Weiheng Testing Instrument Co., Ltd., Shanghai, China) at room temperature (~25 °C). When the stress–strain curve began to decline sharply, the experiment was stopped, and the data were recorded. The sample compression test parameters were set as follows: compression speed at 5 mm/min, repeated three times. A fresh specimen was used for each test. The slope of the measured stress–strain curve was set at a strain lower than 5%, which was recorded as the compressive modulus of the scaffold.

### 4.6. Swelling Termination

In order to avoid structural changes caused by high temperature drying of silk protein samples, the bioscaffold was balanced overnight at 37 °C and 50% relative humidity in a constant temperature and humidity chamber and weighed as *M*_0_. The balanced bioscaffold was immersed in PBS and placed in 37 °C water for absorption and swelling. The scaffold was removed after 1 h, and the water on the surface of the sample was absorbed with filter paper and weighed as *M*_n_. The swelling rate was calculated according to the following formula: swelling rate (%) = *M*_n_/*M*_0_ × 100%. Three samples were repeated for each sample, and the average value and standard deviation were calculated.

### 4.7. Determination of Pore Size and Porosity

The diameters of ~20–50 micro holes in the scaffolds of each group were measured in SEM images by Image J software, and the average pore size was calculated. The porosity of the sample was calculated by the liquid displacement method. The procedure was carried out as follows: the frozen and dried biological scaffold was immersed in a 10 mL cylinder with volume (V_1_), and after 2 h, the total volume of solution (V_2_) was recorded; the sample bracket was removed, and the remaining liquid volume (V_3_) was recorded; the porosity of the scaffold was calculated using the following formula, and repeated three times for each group:Porosity (%) = (V_1_ − V_3_)/(V_2_ − V_3_) × 100%(1)

### 4.8. Structure Analysis Methods

Thermal properties, infrared spectrum, and X-ray diffraction patterns of various powdered biosamples were determined using the methods and corresponding analytical conditions described in the authors’ recent report [57]. Thermal analysis of the sample powder (8.0 mg) was carried out with an SDT2960 instrument at a heating rate of 10 °C/min with a temperature range of 25–800 °C. The transmission spectra (4000–400 cm^−1^) of the sample powder (1.0 mg) were obtained using a Nicolet 6700 FTIR spectrometer with a scanning number of 16 and at a resolution of 4 cm^−1^. X-ray diffraction analysis of the sample powders was performed with a D8 Advance X-ray diffractometer recording pattern in the angle range of 5–50° (scanning condition: 0.03 °/step, 12.5 °/min).

### 4.9. SEM

When the SP scaffolds reached swelling equilibrium, they were frozen at –80 °C, then were freeze-dried, and thin sheets of the scaffolds were taken as samples. The surface and cross-section morphologies of the samples were observed with scanning electron microscopy (Hitachi, Regulus 8230, Tokyo, Japan) using gold spray for 70 s. SEM test acceleration voltage was set to 15 kV.

### 4.10. Enzymatic Hydrolysis

A neutral protease (50 U/mg, Aladdin Ltd Co., Shanghai, China) solution with a concentration of 10 U/mL was prepared with PBS, filtered and sterilized, and placed into aseptic centrifuge tubes. The dry weight of the scaffold and the mass volume ratio of protease solution was 50 mg/mL at 37 °C in a biochemical incubator. Fresh enzyme solution was replaced once a day. Biological scaffold residue was dried and weighed after regular removal, and the percentage of residue after enzymatic hydrolysis of the biological scaffold was calculated.

### 4.11. Drug Release Rate Measurement

Bovine insulin solution (27 U/L) dissolved in PBS was used to filter, sterilize, shake, and mix with SP bioink at the volume ratio of 1:9. It was subsequently injected into the injection syringe of an Allevi 1 Printer (NY, USA) to print a cylindrical porous biological scaffold with a diameter of 12 mm and a height of 8 mm. The drug-containing scaffold was placed in a sterile centrifuge tube, 4 mL sterile PBS was dripped in, and 37 mL PBS at 37 °C was added to the biochemical incubator. PBS once replaced once a day. The content of insulin released into the PBS solution was determined by enzyme-linked immunosorbent assay (ELISA), and the release rate of insulin was calculated.

### 4.12. Cell Culture and Viability Determination

According to the method previously reported by the authors [57], the culture medium of mouse fibroblast L-929 was DMEM high glucose medium containing 10% fetal bovine serum and 1% penicillin at 37 °C and a 5% concentration of CO_2_. In order to study the effect of bioprinting ink on the viability of L-929 cells, CCK-8 was used to measure daily the cell viability of various biological scaffolds. The amount of 10 µL of CCK-8 solution was added to each well of the 48-well plate and incubated again in the incubator for 3 h. In order to reduce the experimental error and avoid the influence of SP scaffolds on the OD value, the cell culture medium of each group was transferred to a new 48-well plate, and the OD value of each well at 450 nm wavelength was recorded, with five parallel plates in each group.

### 4.13. Data Analysis

The experimental data were calculated by Origin 10.0 statistical software. The results are presented as the mean ± SD (standard deviation). Statistical analysis for experimental data were carried out using a one-way analysis of variance (ANOVA) analysis. A *p*-value < 0.05 was considered to be significant.

## 5. Conclusions

This study reports the development of a green method for degumming both SF fibers and whole sericin samples, as well as a process for obtaining inner sericin samples using a high-temperature and high-pressure method. These three proteins were grafted with glycidyl methacrylate to obtain three different grafted silk proteins: G-SF, G-WS, and G-ILS. A variety of SP inks was prepared with PBS and a photoinitiator and used to bioprint a variety of SF scaffolds and SP composite scaffolds. The experimental results showed that the SF grafting reaction obtained by degumming 2.5% GMA and 0.05% Ca(OH)_2_ for 10 min resulted in the most optimal SP composite scaffold, incorporating grafted whole sericin that significantly increased the maximum compressive strain. The compressive strength and maximum compressive strain of the SP composite scaffolds containing G-ILS were also increased. In addition, both the G-WS and G-ILS sericins were able to increase average pore size and porosity. The three SP scaffolds, namely SF alone, SF/WS (10%), and SF/ILS (20%) were composed of mainly β-sheet structures, which delayed the hydrolysis of neutral protease, as well as the rate of insulin release after 5 days, and reached 52% and 62%, respectively. Fluorescence microscopy observation showed that L-929 cells adhered and grew well on the SP scaffolds and demonstrated good cell compatibility.

## Figures and Tables

**Figure 1 ijms-23-00910-f001:**
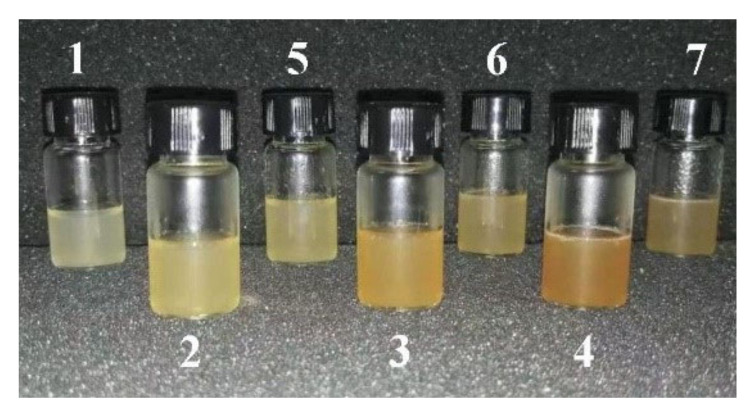
SF ink and SP composite ink prepared by G-SF, G-WS, and G-ILS. The number of SF ink and SP composite ink in the figure are (1) G-SF, (2) G-SF/WS (10%), (3) G-SF/WS (20%), (4) G-SF/WS (30%), (5) G-SF/ILS (10%), (6) G-SF/ILS (20%), and (7) G-SF/ILS (30%).

**Figure 2 ijms-23-00910-f002:**
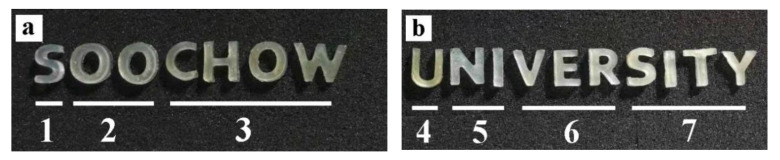
The 3D composite scaffold based on silk protein. The number of SF scaffolds and SP composite scaffolds printed with SP ink are shown in (**a**): (1) G-SF, (2) G-SF/WS (10%), (3) G-SF/WS (20%); (**b**): (4) G-SF/WS (30%), (5) G-SF/ILS (10%), (6) G-SF/ILS (20%), and (7) G-SF/ILS (30%).

**Figure 3 ijms-23-00910-f003:**
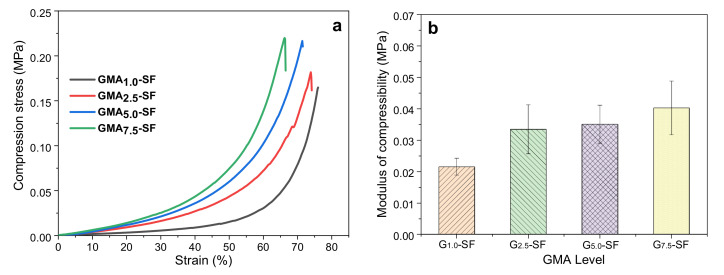
Effect of GMA content on mechanical properties of alone SF scaffolds: (**a**) stress–strain curve; (**b**) compressive modulus.

**Figure 4 ijms-23-00910-f004:**
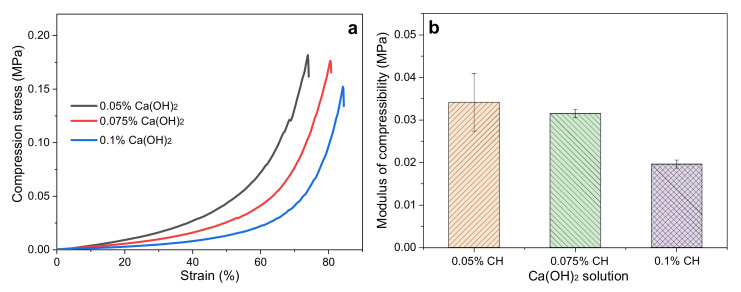
Effect of Ca(OH)_2_ concentration on the mechanical properties of the subsequently synthesized pure SF scaffolds: (**a**) stress–strain curve; (**b**) compressive modulus.

**Figure 5 ijms-23-00910-f005:**
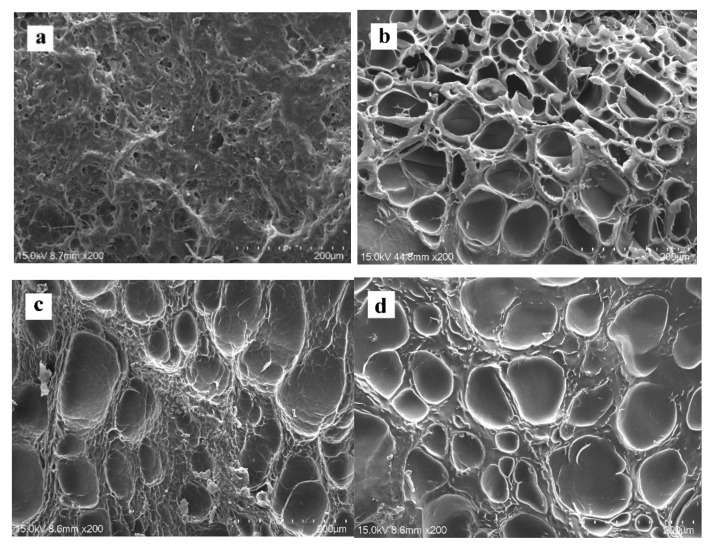
Surface and cross-sectional microstructure of SP composite scaffolds (×200): (**a**) the surface of pure SF scaffold; (**b**–**d**) cross section structures of pure SF scaffold, SF/WS (10%) and SF/ILS (20%) composite 3D, respectively; All scales are 200 µm.

**Figure 6 ijms-23-00910-f006:**
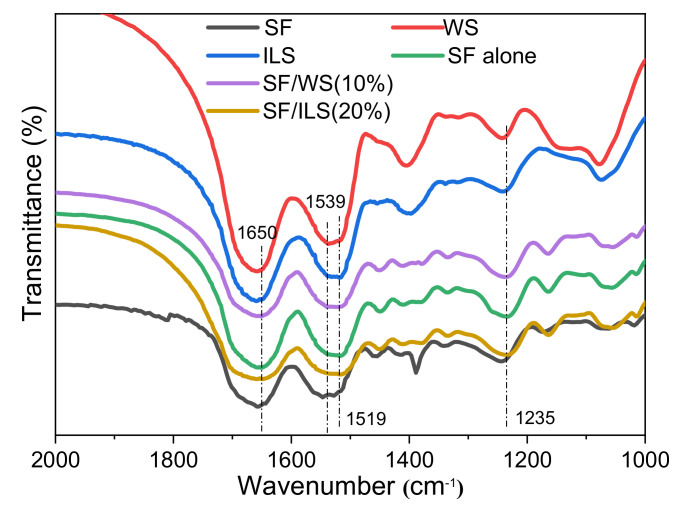
FT-IR spectra of various SP scaffolds.

**Figure 7 ijms-23-00910-f007:**
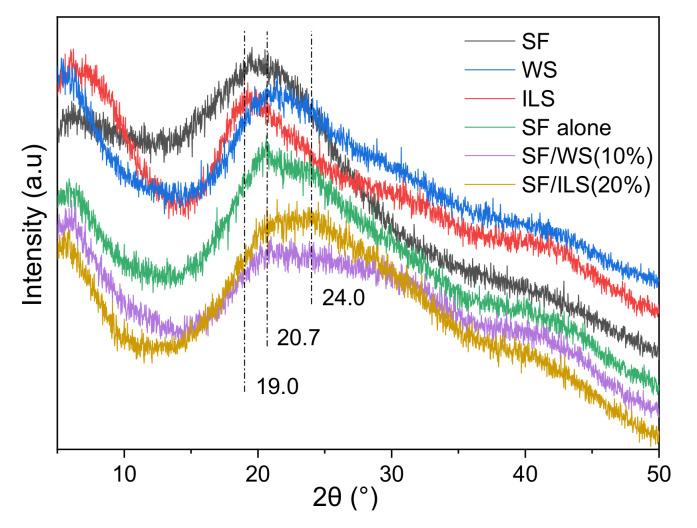
X-diffraction patterns of SP powder and SF scaffold and their SP composite scaffolds.

**Figure 8 ijms-23-00910-f008:**
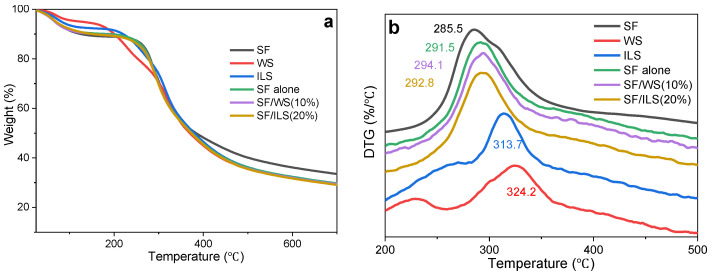
Thermal analysis TG (**a**) and DTG (**b**) of SF, WS, and ILS and pure SF scaffolds and SP composite scaffolds.

**Figure 9 ijms-23-00910-f009:**
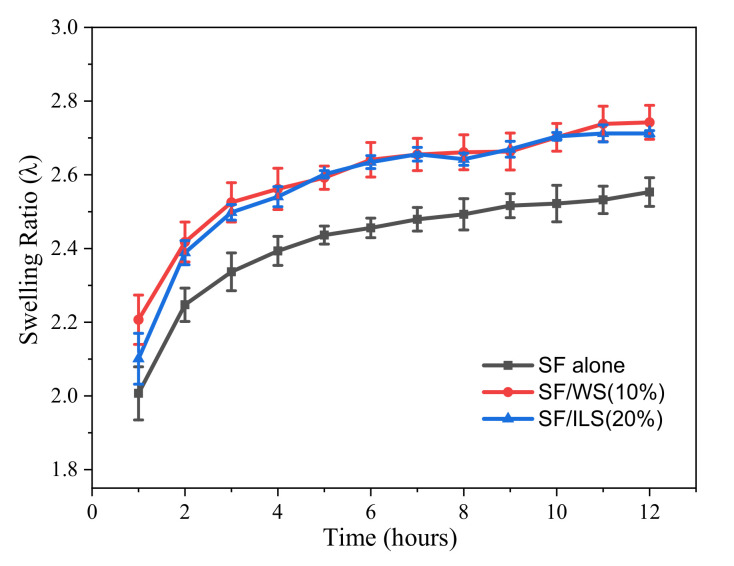
Swelling behavior of pure SF scaffold and SP composite scaffolds. The SF alone, GSF/WS (10%), and SF/ILS (20%) represent the grafted SF scaffolds and the two SP composite scaffolds mixed with 10% G-WS and 20% G-ILS, respectively; *n* = 3.

**Figure 10 ijms-23-00910-f010:**
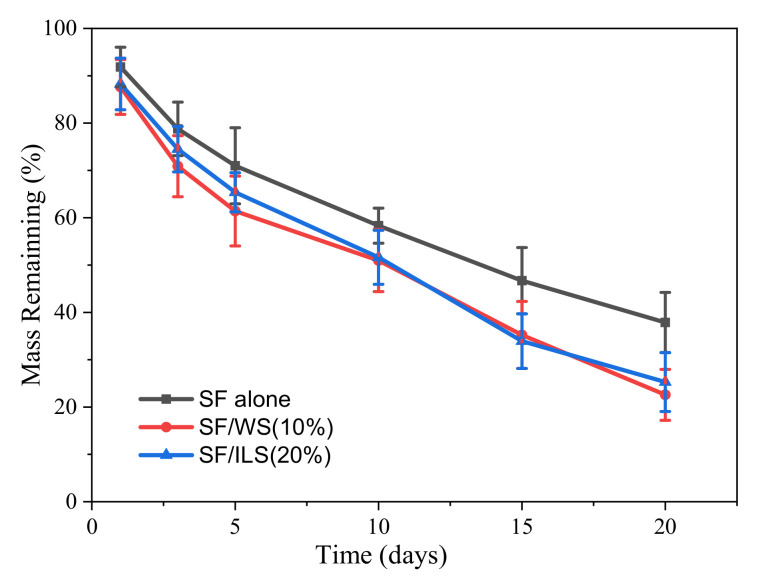
Enzymatic hydrolysis behavior of pure SF scaffold and SP composite scaffolds. The SF alone, SF/WS (10%), and SF/ILS (20%) represent the grafted SF scaffold and the two SP composite scaffolds mixed with 10% G-WS and 20% G-ILS, respectively; *n* = 3.

**Figure 11 ijms-23-00910-f011:**
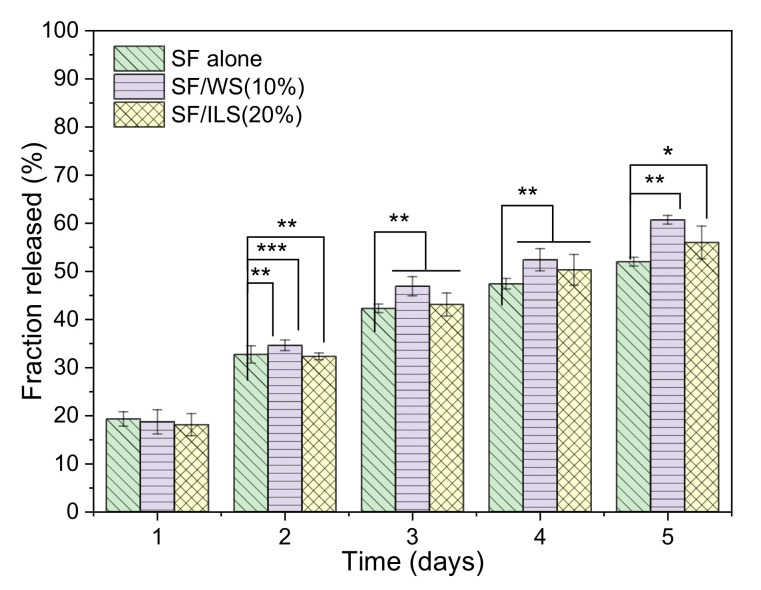
Histogram of bovine insulin release of pure SF and SP composite scaffolds. G-SF, SF/WS (10%), SF/ILS (20%) represent the graft SF scaffold and the incorporation of 10% G-WS and 20% G-ILS, respectively; *n* = 3. * Represents *p* < 0.05, vs. SF alone; ** Represents *p* < 0.01, vs. SF alone; *** representing *p* < 0.001, vs. SF alone.

**Figure 12 ijms-23-00910-f012:**
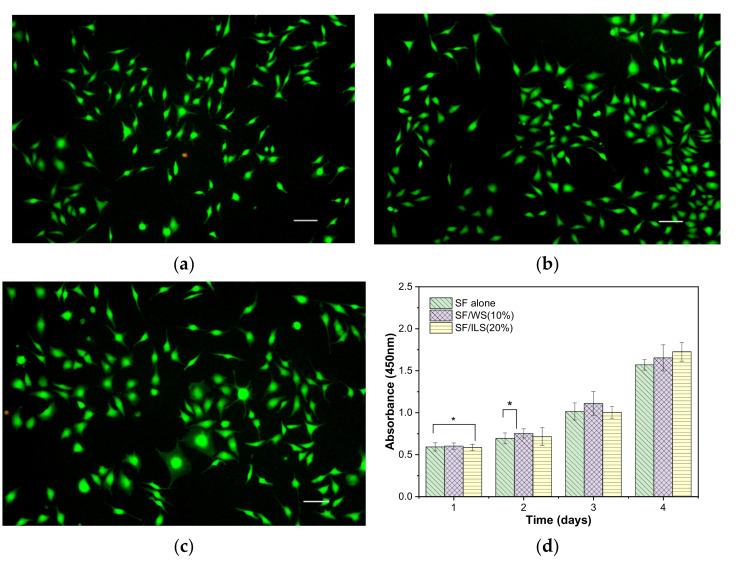
The proliferation of mouse fibroblast L-929 on the SP scaffolds. Microscopic fluorescence photographs (200×) of L-929 cells on day 4 of the three SP scaffolds: SF alone (**a**); SF/WS (10%) (**b**); SF/ILS (20%) (**c**); and their proliferation curves on days 1 to 4 (**d**). All scales are 200 µm; * represents *p* < 0.05; the difference was significant.

**Table 1 ijms-23-00910-t001:** Composition of SF ink and SP composite ink (total volume 20 mL).

No.	Composite Bioinks	G-SF (g)	G-WS (g)	G-ILS (g)	I2959 (g)
1	G-SF	5.00	-	-	0.10
2	G-SF/WS (10%)	4.50	0.50	-	0.10
3	G-SF/WS (20%)	4.00	1.00	-	0.10
4	G-SF/WS (30%)	3.50	1.50	-	0.10
5	G-SF/ILS (10%)	4.50	-	0.50	0.10
6	G-SF/ILS (20%)	4.00	-	1.00	0.10
7	G-SF/ILS (30%)	3.50	-	1.50	0.10

**Table 2 ijms-23-00910-t002:** Effect of silk fiber degumming time on the mechanical properties of pure SF scaffolds.

Degumming Time (min)	Compression Strength (MPa)	Maximum Compression Strain (%)	Compression Modulus (MPa)
10	0.20 ± 0.01	70.27 ± 1.96	0.055 ± 0.003
20	0.20 ± 0.02	73.48 ± 5.10	0.039 ± 0.011
30	0.22 ± 0.06	73.25 ± 8.38	0.030 ± 0.011
40	0.21 ± 0.02	74.47 ± 1.16	0.037 ± 0.001
50	0.17 ± 0.01	71.01 ± 1.49	0.036 ± 0.002

Note: Data are expressed as mean ± SD, repeated three times per group.

**Table 3 ijms-23-00910-t003:** Compression strength, maximum compression strain, and compression modulus of SP composite 3D.

3D Scaffolds	Compression Strength (MPa)	Maximum Compression Strain (%)	Compression Modulus (MPa)
SF alone	0.20 ± 0.01	70.27 ± 1.96	0.055 ± 0.003
SF/WS (10%)	0.24 ± 0.04	78.37 ± 1.00	0.043 ± 0.003 ***
SF/WS (20%)	0.21 ± 0.06	77.80 ± 1.59	0.029 ± 0.010 **
SF/WS (30%)	0.11 ± 0.05	76.28 ± 0.66	0.028 ± 0.003 ***
SF/ILS (10%)	0.23 ± 0.05	78.20 ± 1.12	0.034 ± 0.001 ***
SF/ILS (20%)	0.29 ± 0.03	75.11 ± 6.28	0.042 ± 0.007 **
SF/ILS (30%)	0.25 ± 0.06	73.37 ± 2.07	0.046 ± 0.005 *

The data are expressed as mean ± SD, repeated three times in each group, * representing the difference compared with the pure SF scaffold * representing *p* < 0.05, the difference was significant; ** representing *p* < 0.01, the difference was very significant, *** representing *p* < 0.001, the difference was very significant.

**Table 4 ijms-23-00910-t004:** Average pore size and porosity of SP composite scaffolds.

3D Scaffolds	Average Aperture (µm)	(%) Porosity
SF alone	70.7 ± 18.2	79.7 ± 4.6
SF/WS (10%)	71.2 ± 22.4	83.0 ± 3.4
SF/ILS (20%)	74.9 ± 19.3	82.6 ± 5.3

The data are expressed as mean ± SD, each group is repeated three times, and the average pore size is 20~50 micropores.

## Data Availability

Code and material; The datasets used and/or analyzed during the current study as well as analysis scripts are available from the corresponding author on reasonable request.

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
