# Peer review of "Silk Protein Composite Bioinks and Their 3D Scaffolds and In Vitro Characterization"

_ijms, 2022, doi:10.3390/ijms23020910_

Round 1

Reviewer 1 Report

The manuscript presents a high research regarding silk polymers usage within 3D printing formulations. The authors obtained printable bioinks with potential to develop SF scaffolds. However, the manuscript has several approaches to be addressed or improved:

The abstract must be rephrased in order to clearly expose the research aim and research steps. For exemple (but not only):

  • The phrase The resulting whole sericin (WS) can not only be recycled, but completely degummed silk fibroin (SF) is also obtained in this process is unclear.
  • What represents I2959 and what is the purpose of using it?

In the case of Introduction, I suggest several approches: Line 42 - The polymer has a molecular mass of more than 410 kDa and consists of a heavy chain, light chain, 43 and glycoprotein P25 in a 6:6:1 molar ratio. Only silk fibroin has a such molecular mass and two types of chains, not the entire silk. Furthermore, there is ratio between the fibroin, sericin and P25. Line 45 - I would replace the term crystal with crystallinity.

Please explain why the treatment with calcium hydroxide is a green one. The treatment with calcium carbonate/bicarbonate is not considered a green one (salt treatments).

In the aim please highlight the role of grafting (what type of grafting agent)

The domain 2.1.Materials is not complete

2.3. Grafting methods - What represents G? Is the grafting agent (what type; which is the reaction pathaway/mechanism)?

Results

3.2. Mechanical properties

In term of mechanical properties the addition of silk sericin could not bring the desired reinforcement. This fact is normal because silk sericin has a low crystallinity degree with respect to silk fibroin. The mechanical can be improved by ranging the grafting and crosslinking degree. I suggest to calculate the modulus at a strain lower that 5% (probably there is a more liniar region with higher modulus values)

The FTIR investigation can be used to prove the success of the grafting reaction. Furthermore, the FTIR spectra must reveal the differences between samples (SF alone has only SF and it does not contain sericin; WS/IF has only the specific type of sericin, etc.)

Reviewer 2 Report

This manuscript by Li et al. describes the preparation and characterisation of bio-inks based on silk proteins, and their use to produce scaffolds.  This is an important subject, worthy of investigation and reporting.  In my opinion, however, the present manuscript is not acceptable for publication, due to a large number of issues, which are listed below.

Therefore, I recommend extensive revision, before reconsidering it for publication.

1) L42-44:  The authors state that 'the polymer has a molecular mass of more than 410 kDa and consists of a heavy chain, light chain, and glycoprotein P25 in a 6:6:1 molar ratio'.  This is rather confusing, however.  The silk fibroin, composed of the disulphide-linked 'heavy' and 'light' chains has a formula weight around 410 kDa.  Hence, the suggested 6:6:1 complex has a molecular mass in excess of 2.4 MDa.

But it is not clear that this complex persists beyond protein storage in the posterior silk gland.  Recent studies (small-angle scattering, rheology and modelling) suggest that the fibroin is present as a dissolved polymer of around 410 kDa in the middle silk gland and beyond.

I suggest the authors should clarify this statement in the text.

2) L72: The authors describe sericin as '...a globular protein with a molecular weight of 14 to 314 kDa'.  This is a very broad range for a single protein.  And, actually, literature suggests that several sericin proteins are present in natural silk.  For example, see:
Biosci. Biotechnol. Biochem. 2002, 66(12) 2715-2718;
Insect Biochem. Mol. Biol. 2010, 40, 339-344; doi:10.1016/j.ibmb.2010.02.010

The authors should clarify their statement in the text and provide suitable supporting references.

3) L76 and L504: It is not clear what is meant by 'whitening'.  What is being whitened?  The authors should clarify this in their text, please.

4) L85: Please define these abbreviations where they first appear in the main text.

5) L87-88, L107-108, L479: How is this method (using Ca(OH)2 solution) more 'green' than using soap or other surfactants?  For example, see:
Text. Res. J. 2015, 86(3) 275-287; https://doi.org/10.1177/0040517515586166

The authors should explain, please.

6) L114: Please define what G is, where it first appears in the text.  I presume it is not the same as the G mentioned at line 44.

7) L130: What are I2959 and PBS.  Please define abbreviations where they first appear in the text.

8) L143: I presume a fresh specimen was used for each test.  Please clarify in the text.

9) L147: The authors state that the scaffold samples were dried overnight at 37°C.  Did that relatively low temperature achieve a dry scaffold?  How was it checked?  If M0 is incorrect, it will affect subsequent swelling calculations.

10) L154-155:  The authors state that the hole diameters were measured using ImageJ.  But what imaging method was used: e.g. optical microscopy, SEM or X-ray microtomography?  The authors should specify that in the text.

11) L163: The authors state that thermal properties, infrared spectra and X-ray diffraction pattens were measured.  But how were these analyses performed?  The authors should include how the samples were prepared and the quantities used, if that might affect the results.  Note: some of these important details (such as weights used for thermal analysis, how the samples were presented for IR and angular scanning rate for XRD) are absent from the authors' previous publication [ref. 35].

By the way, X-ray diffraction does not produce a spectrum.  A spectrum relates intensity to different wavelengths (e.g. consider IR spectroscopy, which does give a spectrum.)  Typically, in X-ray diffraction, a single wavelength is used, and the radiation is diffracted to different angles by the material under investigation - but the wavelength remains constant.  Please correct the text accordingly.

12) L173: The authors should specify which protease was used and how the solution was prepared.

13) L190: This reference [xxxvxxxiv] does not appear to be a correct Roman numeral.  I suggest coming more up to date with modern numerals (i.e. Hindu-Arabic, from around the 6th or 7th century) would be better. 

14) L209: The authors state that the 'proportion increased'.  But 'proportion' of what?  The authors should clarify that in the text, please.

15) L212-213: The authors state that the 'the color of whole sericin powder is darker than that of inner sericin powder'.  Does that indicate decomposition or contamination?  Typical Bombyx mori silk fibres without degumming are white - suggesting that the natural sericin itself is white.  Moreover, there appears to be nothing about the protein structures of the sericins to suggest they should be coloured.

Can the authors comment, please.

16) L239-240: I think this is the wrong word.  A stent is a small mesh tube that open passages within the body.  Was that really the form of the test specimen?  The authors should check and correct, if necessary, please.

17) L254: The authors should indicate the temperature (was this 60°C) and time used for the degumming, please.

18) I suspect that the results given in tables 2 anbd 3 are quoted to an overly optimistic level of precision.  For example, how were the thickness and diameter of the test specimens determined?  Were those measurements really sufficiently precise to quote the compressive strength, strain or modulus to 3 decimal places?  This issue is particularly obvious in the strain results, where the SD was larger than 1 % in all cases.  Consequently, the decimal places in the average strain results are meaningless.

The authors should revise these tables to show the results in a more realistic form.

19) L316-317: How was the ideal pore size determined?  Is it from other work?  If so, a reference is required.

20) L334-341: Some references are required to support these infrared peak assignments.

Also, how were the spectra obtained - was it some form of transmission measurement or using attenuated total reflectance (ATR)?

The authors should be aware that the use of ATR can produce significant differences from a transmission spectrum, including apparent shifts in peak position; e.g. see:
https://doi.org/10.1021/jp101763y
https://doi.org/10.1366/0003702001948222

21) L337-339: With regards to the thermal analysis, can the authors suggest what these 'low-temperature volatile components' might be, please?  Are they likely to be contaminants in the scaffold samples, or decomposition producs?  

Also (L382-383) The presence of over 30 % residual weight above 400°C suggests a considerable fraction of inorganic residue, after the organic material (protein) had degraded.  Can the authors comment, please?

Thirdly, (L391), can the authors clarify where N-N bonds are likely to be in the scaffolds, please?  

22) L437:  Fig. 12 shows histograms, not a curve.

There were also some grammatical errors:

Abstract, L17-18: 'The thermal decomposition temperature of the SF/WS(10%) and SF/ILS(20%) scaffolds, mainly composed of a ß-sheet structures, was 3°C and 2°C higher....' should be 'The thermal decomposition temperatures of the SF/WS(10%) and SF/ILS(20%) scaffolds, mainly composed of a ß-sheet structures, were 3°C and 2°C higher....'  (Two materials mentioned, with 2 different decomposition temperatures, so plural.)

Abstract L20: '...bovine insulin release rate reached 61% and 56% after 5 days.' should be '...bovine insulin release rates reached 61% and 56% after 5 days.'  (i.e. Plural.)

L88: 'lyme' (a type of disease caused by a group of spirochaetal bacteria, often carried by infected ticks) should be 'lime' (a rather old-fashioned term for calcium hydroxide) - or, better still, just call it calcium hydroxide.

Round 2

Reviewer 2 Report

I thank the authors for addressing my comments on the previous draft of their manuscript.  In my opinion, those comments have been addressed adequately and I am happy to recommend that the revised manuscript should be accepted for publication.